# Layer-by-Layer Materials for the Fabrication of Devices with Electrochemical Applications

**Eduardo Guzmán** [1,2,*] , **Francisco Ortega** [1,2] **and Ramón G. Rubio** [1]

1  Departamento de Química Física, Facultad de Ciencias Químicas, Universidad Complutense de Madrid, Ciudad Universitaria s/n, 28040 Madrid, Spain; fortega@quim.ucm.es (F.O.); rgrubio@quim.ucm.es (R.G.R.)
2  Instituto Pluridisciplinar, Universidad Complutense de Madrid, Paseo Juan XXIII 1, 28040 Madrid, Spain
*  Correspondence: eduardogs@quim.ucm.es; Tel.: +34-91-394-4107

**Abstract:** The construction of nanostructured materials for their application in electrochemical processes, e.g., energy storage and conversion, or sensing, has undergone a spectacular development over the last decades as a consequence of their unique properties in comparison to those of their bulk counterparts, e.g., large surface area and facilitated charge/mass transport pathways. This has driven strong research on the optimization of nanostructured materials for the fabrication of electrochemical devices, which demands techniques allowing the assembly of hybrid materials with well-controlled structures and properties. The Layer-by-Layer (LbL) method is well suited for fulfilling the requirements associated with the fabrication of devices for electrochemical applications, enabling the fabrication of nanomaterials with tunable properties that can be exploited as candidates for their application in fuel cells, batteries, electrochromic devices, solar cells, and sensors. This review provides an updated discussion of some of the most recent advances on the application of the LbL method for the fabrication of nanomaterials that can be exploited in the design of novel electrochemical devices.

**Keywords:** capacitors; electrochemistry; electrodes; energy storage; Layer-by-Layer; miniaturization; thin film

## 1. Introduction

The miniaturization of high-tech electronic devices demands stable microscale power sources, e.g., microbatteries, to guarantee their portability. This miniature power sources should ensure high power, high energy, and a long cycle life, maintaining a low production cost [1], which emerges as a strong drawback for their fabrication by using only traditional components, e.g., inorganic/metal compounds, carbonaceous substances, and petroleum-derived hydrocarbon chemicals [2]. Furthermore, the fabrication of microbatteries and electrochemical capacitors requires the fabrication of electrodes of well-controlled thicknesses, providing high energy and power, which is not always easy using traditional approaches [3,4]. A possible solution for the fabrication of efficient miniaturized electrochemical devices is to take advantage of the versatility and modularity of the Layer-by-Layer (LbL) method for the fabrication of hierarchical nanomaterials [5,6].

The LbL method was initially introduced by Decher et al. [7] in the early 1990s for the fabrication of nanostructured materials using a methodology based on the alternate deposition of polyelectrolytes bearing opposite charges [7,8]. However, it underwent fast development, and in the current state of development it allows the fabrication of nanomaterials by combining any type of mutual interacting species which can be assembled through a broad range of types of interactions, e.g., electrostatics, hydrogen bonding, charge transfer interactions, molecular recognition, coordination interactions, chiral recognition, host–guest interactions, π–π interactions, biospecific interactions, sol–gel reactions, or even covalent bond ("click chemistry" reactions) [9]. This expands the range of materials that can be used as building blocks of LbL material beyond polyelectrolytes and bolaamphiphiles [7,10],

allowing the fabrication of LbL materials with a broad range of molecules and colloidal nano-objects, irrespectively of their charge density. In fact, the LbL is not limited to the assembly of charge block, and the list of materials that can be assembled for building LbL materials includes, among other components, synthetic oppositely charged polyelectrolytes; synthetic neutral polymers; colloidal particles and nano-objects; biomolecules; dyes; viruses; and even, in some cases, molecular species [11]. On the other hand, the assembly process of LbL films, as well as their structures and properties, can be easily tuned by modifying different physico-chemical parameters of the layering solutions that modify the intricate balance of interactions between the assembly block (pH, ionic strength, temperature, polarity, concentration, building blocks charge density) or the type of substrate used as template of the assembly process [12–30]. This is important because the LbL method is a template-assisted method which can be used for the assembly of nanostructured materials by using as template any solvent accesible surface. This allows the fabrication of LbL materials using templates that differ from traditional macroscopic solid charged substrates, and hence it is possible to find examples of LbL films supported on colloidal micro- and nanoparticles, liposomes or vesicles, micelles, fluid interfaces (floating multilayers), emulsion droplets, or even cells [29,31–39]. Therefore, the LbL method offers a broad range of possibilities for the fabrication of macromolecular devices by the combination of a broad range of functional building blocks, allowing the fabrication of a broad range of materials, ranging from flat films to nano-/micro-capsules, and from hierarchical multicapsules to onion-like structures, sponges, membranes, or nanotubes [40–42]. Figure 1 shows a scheme highlighting some of the most relevant features of the LbL method.

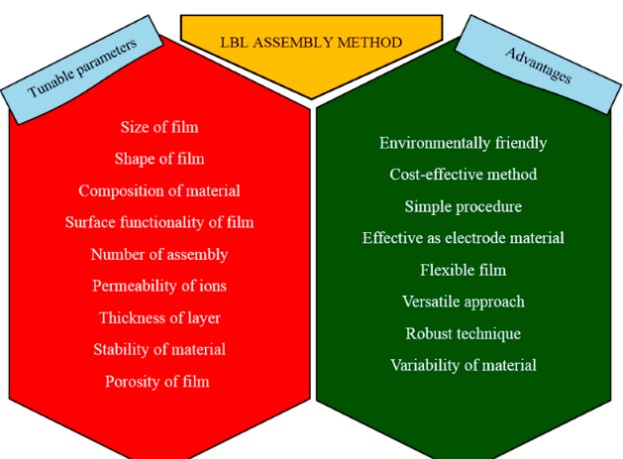

**Figure 1.** Scheme of the most relevant features of the LbL method for the assembly of nanostructures materials. Reprinted from Kulandaivalu et al. [42], with permission under Open access CC BY 4.0 license, https://creativecommons.org/licenses/by/4.0/ (accessed on 9 April 2022).

The LbL method opens new avenues for the fabrication of nanostructured hybrid materials for the design of novel electrochemical devices. Indeed, the possibility of manipulating functional building blocks at the molecular level offers important advantages for modulating the performance of the manufactured materials almost at will, which results in a broad range of opportunities to enhance the electrochemical performance of the manufactured devices. For instance, LbL materials can enhance the energy storage characteristics of batteries or improve the sensitivity of biosensors [43]. This has driven the development of the LbL method as a key enabling technology on the fabrication of new hybrid materials in applications associated with electrochemistry [6,44], taking advantage of the possibilities offered by the LbL method for controlling the composition, morphology, and architecture of the assembled nanomaterials. In fact, the LbL allows the fabrication of materials with tunable mechanical, electrical, and chemical properties, which is essential for the fabrication

of electrochemical devices, including batteries, supercapacitors, fuel cells, solar cells, and sensors, which requires dense films with permeability to ions [45–51].

This work provides an updated perspective to the current trends on the use of LbL materials for the fabrication of materials for electrochemical applications, highlighting some of the most recent research efforts in this field. We hope that this review can be used to open new avenues on the fabrication of LbL materials for electrochemical devices.

## 2. Approaches for the Assembly of LbL Materials for Energy Applications

The LbL method is a simple and cost-effective approach for the fabrication of nanostructured layered materials [52,53]. The most common approach used for such purposes relies on the sequential immersion of a substrate into solutions containing materials with complementary functionalities. Thus, it is possible to manufacture layered materials under mild conditions on substrates of different natures [54–56]. Therefore, the LbL method is a controllable solution-based deposition method for the fabrication of thin films, membranes, and stackable 2D nanomaterials [44,54,57,58].

The simplest approach for the assembly of LbL materials is the so-called immersive dip-coating deposition. This methodology can be easily adapted to the coating of both flat and non-flat substrates, enabling, in some cases, the coating of substrates with very complex geometries [7,10,41,53,54,58–61]. However, the implementation of the dip coating deposition, especially at an industrial level, is limited by the long time required for the fabrication of films with optimal quality, and the high amount of material required for the assembly of each single layer, which in turn results in a huge amount of waste after the fabrication procedure. Despite the above drawbacks, the versatility and simplicity of the dip coating deposition methodology has driven its development as an essential route for the assembly of LbL materials, and currently it is possible to find different automatic dipping devices, which simplifies the fabrication of LbL materials, and enables a significant reduction of the time required for the assembly process. For instance, the use of robotic dippers which provide a rotation to the substrate during its immersion in the solution containing the molecules to be deposited on the substrate leads to a reduction of the time required for the assembly of a finalized material by a factor ranging between 3 and 10. Furthermore, the rotation of the substrate allows modulating the thickness of the obtained layers (the higher the rotation velocity the thicker the obtained layers) [41].

The dip-coating deposition methodology relies on the alternate exposure of the substrate to solutions containing the layering compounds, i.e., the molecules that form the layer, including intermediate rinsing steps between the adsorption of consecutive layers [62]. This rinsing step allows removal of the excess of material which is not strongly adsorbed on the material, avoiding the cross-contamination phenomena during the assembly process of the nanomaterial. This is particularly important when the assembly of charge species is concerned because the interaction of oppositely charge molecules in a solution can result in the formation of insoluble complexes, e.g., interpolelectrolyte ones, which can precipitate onto the LbL films, altering the composition, structure, and properties of the assembled nanomaterial [62–64]. Figure 2 presents a sketch of the assembly process of LbL films using a dip-coating deposition methodology.

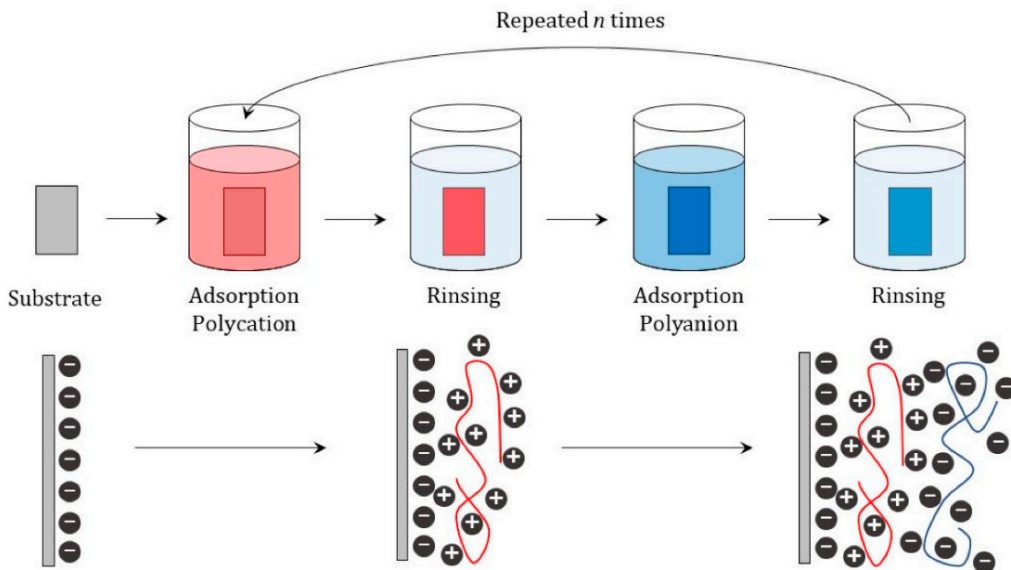

**Figure 2.** Sketch of the assembly of LbL nanomaterials by a dip-coating deposition methodology. The sketch represents the particular case of the assembly of a multilayer by the assembly of polyelectrolytes bearing opposite charges using a negatively charged flat substrate as template, and intermediate rinsing steps. In the bottom panel, the red line corresponds to the polycation layer evidenced by its positive charges, and the blue line corresponds to the polyanion layer evidenced by its negative charges. Reprinted from Mateos-Maroto et al. [64], with permission under Open access CC BY 4.0 license, https://creativecommons.org/licenses/by/4.0/ (accessed on 7 April 2022).

One of the challenges concerning the design of more effective and faster methodologies for the assembly of LbL nanomaterials is to remove the intermediate rinsing steps from the fabrication process. This can be possible by introducing the dewetting approach [58,65]. This relies on the deposition of the layers from solutions containing an organic solvent, e.g., dimethylformamide or dimethylsulfoxide, which upon evaporation can induce a dewetting of the deposited films. Thus, it is possible to obtain a reduction of the time required for the material fabrication by a factor of about 30, enabling a better control of the lateral organization of the adsorbed layers [65]. Another methodology for the increase of the velocity of the assembly process involves the continuous stirring of the solutions, by using a magnetic bar, during the deposition cycles (agitated dip-coating deposition). Thus, it is possible to reduce the time required for depositing each single layer down to 10–20 s, instead of the traditional 15 min, without compromising the final homogeneity of the obtained films [66].

The chemical bath deposition emerges as an alternative to the dip coating deposition, especially when metal based layers are deposited [67]. This procedure is also an inmersion deposition method relying on the layer fabrication by the immersion of the substrates in hot solutions, with bath temperature and inmersion time playing a central role concerning the thicknesses and morphologies of the obtained layers [68]. The use of the chemical bath deposition is many times limited by the formation of precipitates in the electrolyte solutions which may reduce the amount of available material for the assembly of the layers [69]. A modified version of the chemical bath deposition is the fabrication of the layers by following a scheme based in ionic layer adsorption followed by a reaction process [69]. This method relies on the succesive inmersion of the substrates in solutions containing precursor ionic species of those that are expected to form the assembled layers upon a chemical reaction [70].

The long time required for ensuring the diffusion of molecules to the surface, their adsorption, and the final equilibration of the deposited films (around 15 min per the deposition of a single layer) when a dip-coating deposition approach is used has driven the research towards the design of new methodological approaches enabling the fabrication of

LbL materials following a less time consuming procedure. This leads to the exploitation of the deposition assisted by substrate spinning (spin-coating deposition), and the spray assisted deposition as alternatives for a faster fabrication of LbL materials [71]. These approaches offer a way to overcome some of the main limitations associated with the dip-coating deposition, e.g., long time required for manufacturing LbL materials, high amoung of waste, and difficulties concerning its scaling at an industrial level, allowing the fabrication of LbL films with good quality. Figure 3 displays schemes of the general features of the spin-coating and spray-assisted deposition methodologies.

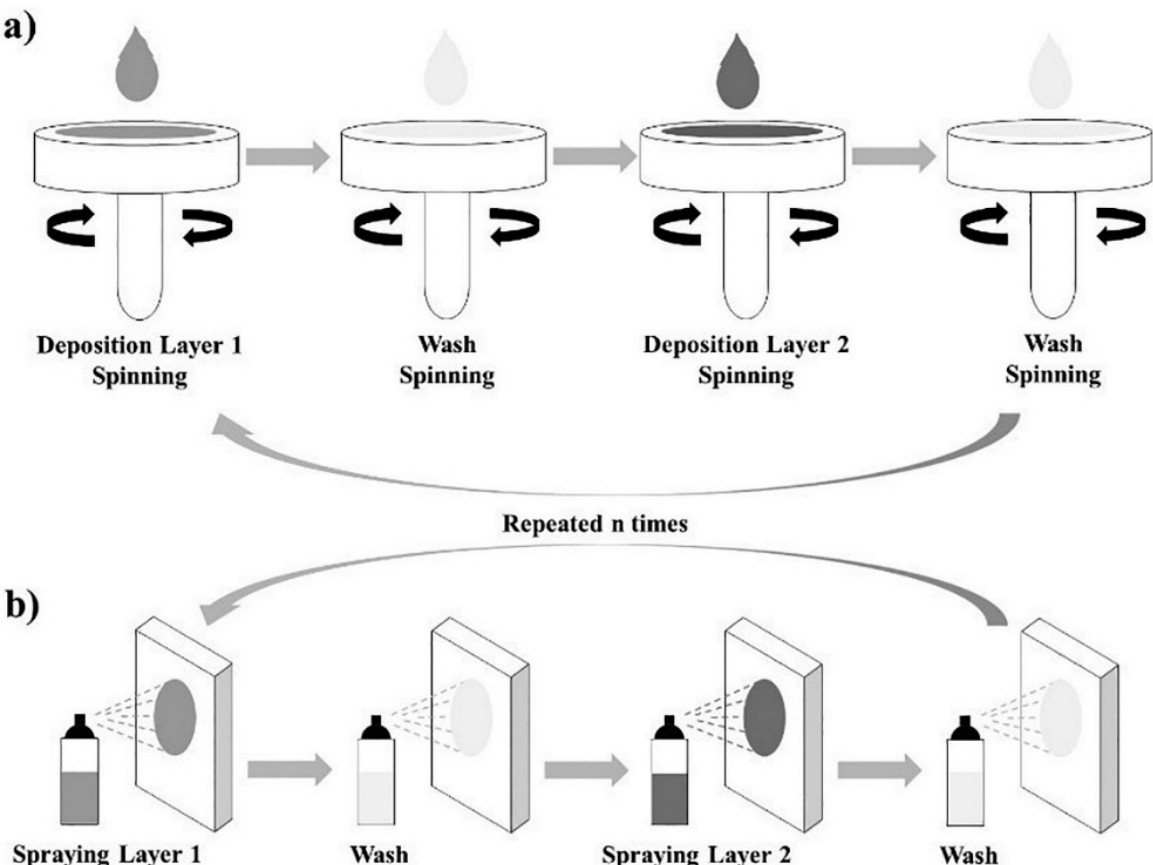

**Figure 3.** Schematic representations of the assembly of LbL nanomaterials by spin-coating (**a**) and spray-assisted (**b**) deposition methodologies. Reprinted with permission from Ref. [41]. Copyright 2020 Elsevier.

The spin-coating deposition relies on the deposition of layers by casting the solution containing the layering compounds on the surface of the substrate which is later subjected to spinning at a constant velocity to ensure the spread of the adsorbing species on the substrate. This makes it possible to obtain a homogeneous film after the complete evaporation of the solvent. Once the layer is completely deposited, it is necessary to rinse the deposited film following a procedure similar to that used for the layer deposition, maintaining the substrate under spinning until the total drying of the deposited material is guaranteed. The assembly of a layered nanomaterial by a spin-coating procedure requires to repeat the deposition and rinsing steps until the desired number of layers are deposited [72,73]. The spin-coating deposition allows obtaining films with a reduced roughness and better organization than those obtained by dip-coating deposition, contributing to the reduction of the degree of interpenetration between adjacent layers. This results from an intricate balance between different types of interactions, including electrostatic and hydrodynamic ones (centrifugal, air shear, and viscous forces), which contribute to the adsorption and rearrangements of the adsorbing species, and the desorption of weakly bound molecules

and the dehydration of the layers [73]. The complex interplay between the different contributions allows the assembly of LbL materiales in a shorter time-scale than when dip-coating deposition is used. Furthermore, the layers obtained following a spin-coating deposition procedure present smaller thickness than those obtained by dip-coating, with the spinning velocity playing a central role in controlling the layer thickness (the higher the spinning velocity the thinner the obtained films), allowing a significant waste reduction [73]. Nevertheless, spin-coating deposition presents a very important drawback related to the low volatility of the water which is the common solvent in LbL deposition, which restricts the applications of the spin-coating deposition of LbL materials to the laboratory scale. It should be noted that the use of spin-coating deposition for the fabrication of LbL materials requires to control carefully both the spinning velocity and the concentration of the layering solutions. Thus, it is possible to tune both the structure and properties of the obtained films [74].

The rotation of the substrate can also be exploited for the fabrication of LbL materials under the application of high-gravity fields for hastening the diffusion of the layering species. Thus, the time needed to attain the equilibrium conditions can be reduced. The use of this methodological approach allows manufacturing materials with similar quality than those obtained using dip-coating deposition, ensuring a reduction of the time required for the assembly process as a result of the concentration gradients and turbulences emerging during the deposition process [75].

The use of spray-assisted LbL deposition relies on the fabrication of layered nano-materials by the alternate spraying of solutions containing materials of different natures on the surface of a flat substrate, including intermediate rinsing steps between two consecutive deposition cycles [76]. Spray-assisted deposition presents advantages similar to spin-coating deposition, ensuring the fabrication of LbL materials with smaller roughness and thickness than those obtained following a dip-coating procedure [77–80]. This is the result of the simultaneous occurrence of two processes during the spraying: (i) Adsorption and (ii) drainage. The latter is favored when the spraying is performed peperpendicularly to the surface, which contributes to a fast removal of the excess of the sprayed solution. Nevertheles, the gravitational drainage may favor the formation of heterogenous layers, in which material can accumulate close to the solution drips [81,82]. A possible solution to this problem is to use rotating substrates during the deposition process [83]. A very important advantage associated with the use of spray-assisted deposition for the fabrication of LbL materials is the short time scales involved in the assembly process (less than 10 s for the deposition of a single layer may be enough), minimizing the degree of interpenetration between adjacent layers [81]. Furthermore, spray-assisted LbL deposition facilitates a fast fabrication of very homogeneous LbL materials on surfaces with a large surface area [84,85]. Despite the methodological simplicity of the spray assisted method, its effectiveness depends on different parameters, including the spray-substrate distance and the times of spraying and drainage [86]. Furthermore, aspects such as the solution concentrations, the volume and flow of the sprayed solutions, the spraying time and the waiting time between two consecutive steps, whether rinsing steps are included or not, and how long these rinsing steps should be should be carefully analysed for a successful fabrication of LbL material by spraying [81,87–92]. The velocity of the assembly of LbL materials using the spray-assisted deposition method can be increased by the application of a vacuum. Thus, it is possible to minimize the lag times between the different steps of the process [86].

The yield of the fabrication of LbL materials by spray-assisted deposition can be improved using a strategy based on the simultaneous spraying of the different species that form the final material. This is the the so-called simultaneous spray coating of the interacting species approach (SSCIS). The use of this methodology allows a fast fabrication of LbL materials, ensuring a continuous and gradual growth by the direct interaction of the assembled species on the substrate. The thickness is controlled by the spraying time, and as occurs in normal spray assisted deposition the solvent and the excess of non-absorbed

material are removed by drainage, leading to materials with many similarities to those obtained by using the conventional spray-assisted deposition [88,93].

The LbL deposition can be modulated under the application of external fields, e.g., electric or magnetic ones [94,95]. This provides a strategy for obtaining LbL materials with higher density and thickness than those obtained using any other methodology for the deposition of LbL materials [96,97]. The deposition under the application of electric fields commonly relies on the use of electrodes as substrates for the LbL assembly (electrodeposition). Thus, it is possible to stimulate the deposition of the films upon the application of a voltage within the formed electrolytic cell. The usual electrodeposition approach is based on the fabrication of a three-electrode cell with the electrodes immersed in the solution of the species that are going to form the layer, and then an electric current is applied to the cell to drive the deposition process on the working electrode (flexible or solid substrate). Figure 4 presents a sketch of a typical electrodeposition for the assembly of LbL materials. Once the layer is deposited on the substrate, a rinsing of the electrodes is performed before continuing with the deposition of the second layer following a procedure similar to that described above. The steps of electrodeposition and rinsing are repeated until a material with the desired thickness and properties (structure, surface area, and rate of deposition) is obtained [98]. These can be controlled by optimizing the applied voltage and its application time [99]. There are several experimental designs allowing the fabrication of LbL materials of different templates independently of their shapes and sizes [58], having as main advantage that electrodeposition LbL assembly is performed under mild conditions (room temperature, non toxic chemical, and minimum use of reagents) [100].

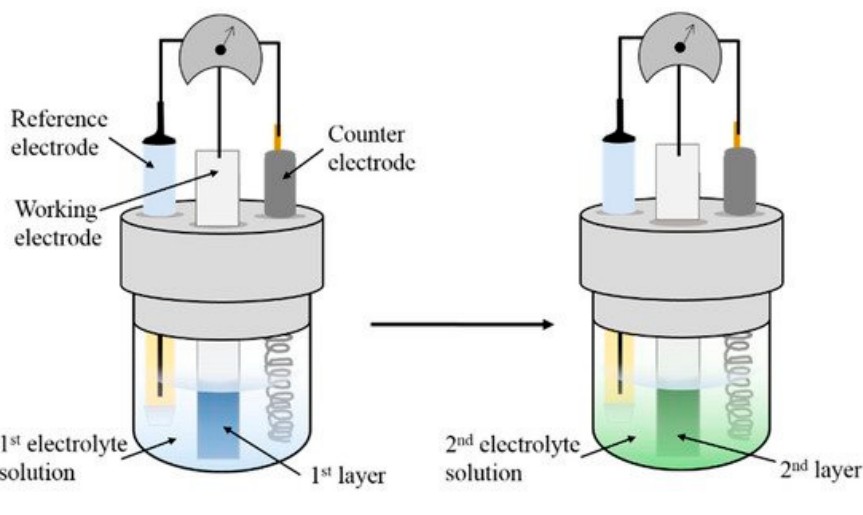

**Figure 4.** Sketch of the simpler procedure for the fabrication by electrodeposition of LbL materials. Reprinted from Kulandaivalu et al. [42], with permission under Open access CC BY 4.0 license, https://creativecommons.org/licenses/by/4.0/ (accessed on 11 April 2022).

The electodeposition of LbL films is possible by exploiting the local effects that can occur on the surface of electrodes, e.g., redox reactions or pH changes. This can lead to a situation in which the local change of pH close to the electrode surface, e.g., the existence of a lower pH close to the anode than in the solution, can stimulate the film deposition [101]. Nevertheless, the use of this local effects for controlling the film deposition is only possible for the deposition of a limited number of layers, because the presence of a film on the electrode surface limits the capacity of the current to penetrate towards the electrode surface, and hence the electrode effect appears screened.

The use of magnetic fields for driving the fabrication of LbL materials is another very powerful alternative [102]. This is possible when the layers are formed by magnetic responsive species. Thus, the assembly of the LbL material is performed following a

common dip-coating deposition procedure, and the magnetic fields is applied between the deposition of adjacent layers for modulating the packing, and consequently the layer thickness [103].

The above discussion has been focused on the most common procedures for the assembly of LbL materials using flat macroscopic substrates as templates. Nevertheless, LbL materials can be also fabricated using substrates of more complex geometries, enabling the fabrication of hierarchical supramolecular systems with a high degree of complexity. The specificities of the assembly procedures of LbL materials using non-flat substrates as templates present a very limited interest for the assembly of materials with applications in electrochemistry, and hence it will not be discussed in this review. A more complete perspective of the different methodologies used for the assembly of LbL materials can be found in previous publications [40,41,58,61,64].

## 3. LbL Nanomaterials in the Fabrication of Electrochemical Devices

The LbL technique has been recently exploited for the fabrication of a large number of functional electrodes with applications in different electrochemical devices. This requires the alternating assembly of different building blocks, e.g., small molecules, metal complexes, macromolecules, nanoparticles, or polymers, to fabricate hybrid materials [42].

It is common the use of up to six different families of components for the fabrication of LbL materials for electrochemical applications [43]. These families include: (i) Conductive polymers; (ii) carbon nanomaterials; (iii) MXene (carbides, nitrides or carbonitrides of transition metal); (iv) pseudocapacitive nanoparticles; (v) layered double hydroxides; and (vi) polyoxometalates. The fabrication of LbL electrochemical devices can combine different building blocks of the above categories, or can include one element of such categories and non-electroactive compounds. Thus, it is possible to fabricate electroactive interfaces with unique electrical, mechanical, and chemical properties, allowing the establishment of synergistic effects between the different components favoring the electrochemical performance of the manufactured devices. Therefore, the LbL method offers many possibilities to fabricate three-dimensional multicomponent architectures with tailored electrical, electrochemical, and transport properties [104]. In the following, a description of different studies of electrodes containing elements of the main categories of electroactive compounds will be presented. However, in some cases, examples of hybrid electrodes, in which two different electroactive compounds belonging to the same family or to a different one are included, will be discussed. In this latter case, the emergence of synergistic interactions is common between the components which reinforces the electrochemical properties of the individual components. For instance, the assembly of layers of two different conductive polymers, namely poly(3,4-ethylenedioxythiophene) and poly(N-methylpyrrole), results in an twofold increase of the electrode capacitance in relation to electrodes consisting only of poly(3,4-ethylenedioxythiophene) [105].

### 3.1. LbL Electrochemical Devices Containing Conductive Polymers

Conductive polymers have been frequently exploited in the fabrication of electrode material for electrochemical devices. The main characteristics of conductive polymers is the alternance of single and double bonds within their backbones. This favors the delocalization of the electronic density along the polymer chain which governs the conductivity of conductive polymers, resulting in special electronic properties. Indeed, conductive polymers present controllable electrical conductivity, low energy optical transition, low ionization potential, and high electron affinities [106,107]. Conductive polymers have been used as layering material in LbL materials aimed at the fabrication of thin film electrodes [42,108].

Yun et al. [108] combined negatively charged MXene nanosheets with positively charged polyaniline nanofibers (PNFs) in the fabrication of LbL hybrid electrodes. The combination of these two types of building blocks allows the fabrication of LbL multilayers with an average bilayer thickness of about 49 nm. These bilayers present an average weight of about 77% polyaniline and 23% MXenes, allowing the fabrication of a sandwich cell of

about 2 µm thickness having maximum areal capacity, energy, and power of 17.6 µA·h/cm², 22.1 µW·h/cm², and 1.5 mW/cm², respectively. Figure 5 shows some characteristics of the multilayers obtained by the combination of alternate layers of MXene and polyaniline nanofibers. This type of LbL system of MXene nanosheets and polyaniline nanofibers emerges as a very good alternative for the fabrication of thin film electrochemical energy storage devices. The design of electrodes for electrochemical energy storage was also explored by Shao et al. [109]. They designed hybrid organic–inorganic electrodes by combining alternate layers of polyaniline and vanadium pentoxide ($V_2O_5$). The obtained electrodes show a better performance than those obtained by the individual components due to the synergistic interaction between the two assembled components. However, the electrochromic behavior of the hybrid electrode is mainly dominated by the polyaniline. On the other hand, the electrochemical response depends on both components, with the molecular weight of the polyaniline playing a very important role in the whole performance of the hybrid electrodes. Furthermore, the thickness of the electrode also affects to the charge-storage ability due to the effect of the differences on the fraction of electrochemically accessible material in the assembled material. This introduces important limitations to the electronic and ionic diffusion, which modifies the capacity of the electrode and its charge–discharge behavior. The best performing electrodes were those assembled with 16 bilayers including low molecular weight polyaniline. These electrodes present a capacity of about 260 mA·h/cm³, and maintain around 80% of their initial capacity after 500 charge–discharge cycles at a current of 20 µA/cm².

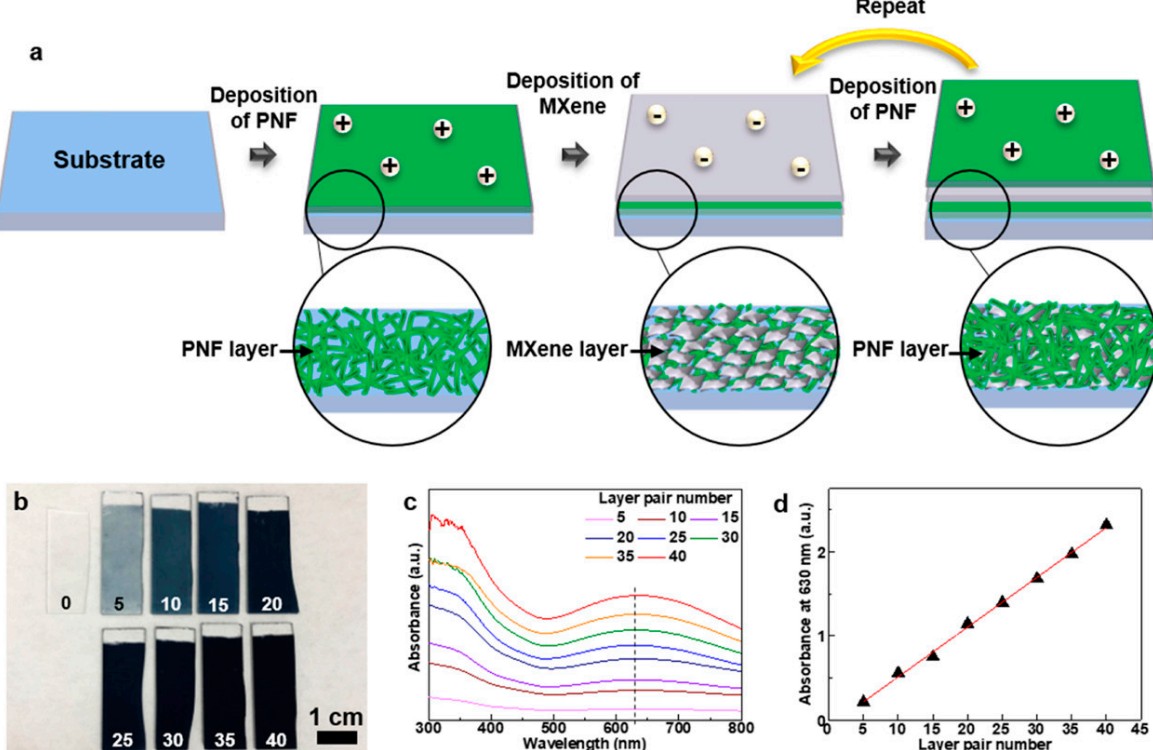

**Figure 5.** (**a**) Sketch of the assembly process of LbL multilayers obtained by alternate deposition of polyaniline nanofibers and MXene nanosheets. (**b**) Image of LbL films with different numbers of bilayers of polyaniline nanofibers and MXene nanosheets deposited on glass. (**c**) UV-Visible spectra for LbL multilayers of polyaniline nanofibers and MXene nanosheets with different numbers of bilayers. (**d**) Absorbance (at 630 nm) dependence on the number of bilayers for LbL multilayers of polyaniline nanofibers and MXene nanosheets. Reprinted with permission from Ref. [108]. Copyright 2019 American Chemical Society.

Shao et al. [110] extended their studies on the fabrication of hybrid electrodes composed of layers of polyaniline and pentoxide of vanadium. They replace conventional polyaniline by polyaniline nanofibers in the electrodes, and found that the obtained electrodes present a highly porous structure with a better performance than electrodes containing conventional polyaniline. In fact, electrodes based on layers of polyaniline nanofibers present higher capacity (three times), specific energy (40 times), and specific power (four times) than the electrodes with conventional polyaniline. Furthermore, the electrochemical performance is strongly dependent on the electrode thickness. The good electrochemical performance of electrodes containing polyaniline nanofibers results from the combination of simultaneous diffusion-limited and non-diffusion-limited redox processes.

Aradilla et al. [105] showed the power of LbL materials formed by alternate layers of poly(3,4-ethylenedioxythiophene) (PEDOT) and poly(N-methylpyrrole) for the preparation of electrodes for the assembly of capacitors. The combination of layers of the two polymers results in an enhanced performance of the obtained electrodes in relation to those including only the individual polymers. This is the result of the differences in topography and morphology of the hybrid systems in relation to the individual components. In particular, LbL materials containing three and five bilayers present a specific capacitance of 50 F/g and 63 F/g, respectively. On the other hand, the high porosity of the multilayers, which favors the ionic diffusion within the polymeric material, results in materials with low internal and low charge transfer resistances. This allows the use of LbL electrodes formed for layers of PEDOT and poly(N-methylpyrrole) as anodes or cathodes in different types of electronic devices. Lee et al. [111] also used layers of PEDOT for the fabrication of capacitors. They combined PEDOT:PSS (poly(3,4-ethylenedioxythiophene)–poly(styrenesulfonate)) layers with polyethyleneimine (PEI) layers for the fabrication of conductive structures supported on cellulose fibers. The conductivity of the obtained material appears in the range $10^{-5}$–$10^{-4}$ S/cm, depending on the number of bilayers, decreasing when the assemblage of the LbL film was performed from solutions of high ionic strength. This latter may result from the swelling of the layers which isolates the PEDOT chains and interrupts the electron transfer between adjacent chains. On the other hand, the assembly from solutions of low ionic strength leads to a densification of the film favoring the contacts between PEDOT:PSS molecules which is the main factor modulating the conductivity of the manufactured structures. A significant improvement of the conductivity was found by replacing the PEI for carbon nanotubes, leading to structures with a conductivity up to 4 orders of magnitudes higher than those found by Lee et al. [111] for LbL films formed by PEDOT:PSS and PEI. The combination of two electrodes in a cell formed by alternated layers of PEDOT:PSS and carbon nanotubes separated by a PEI layer pushes the conductivity of the system up to a maximum value of 20 S/cm, allowing the use of this type of material as capacitors [112].

Hyder et al. [113] also exploited the combination of electrodes formed by a conductive polymer (positively charged doped polyaniline) and carbon nanotubes (multiwalled carbon nanotubes, MWCNT) modified with carboxylic acid groups in the fabrication of LbL electrodes. The LbL film of the electrodes consists in a nanoscale interpenetrating cross-linked network with well-defined nanopores and interconnectivity, which favors a fast electron and ion transport. This ensures an excellent electrochemical performance of the obtained electrodes as demonstrated by their operation in lithium cells. In fact, the LbL film formed by polyaniline and MWCNT layers presents a high volumetric capacitance (around 230 F/cm$^3$) and a high volumetric capacity (around 210 mA·h/cm$^3$). On the other hand, it can deliver simultaneously high power and high energy density (about 220 W·h/L$_{electrode}$ at 100 kW/L$_{electrode}$, with L$_{electrode}$ representing the electrode thickness), allowing their use as positive electrodes in different devices for energy storage and conversion. For instance, the integration of the LbL films on different substrates, e.g., silicon wafers, glass, or flexible substrates, emerges as a very promising approach for the fabrication of a new generation of microbatteries or electrocapacitors, sensors, or actuators. Figure 6 presents a sketch of the assembly process of LbL films of polyaniline and MWCNT layers.

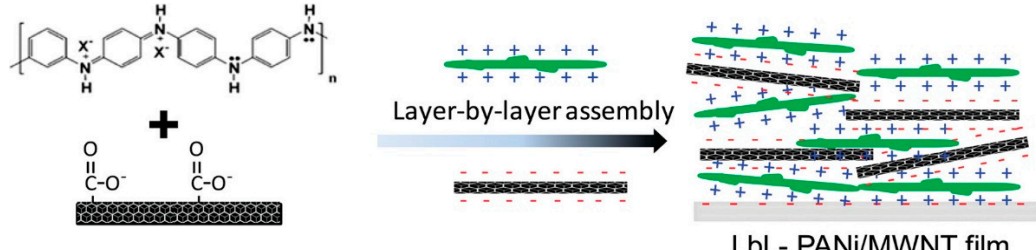

**Figure 6.** Sketch of the assembly process of LbL multilayers obtained by the alternate deposition of polyaniline with positive charges and MWCNT modified with carboxylic groups. Reprinted with permission from Ref. [113]. Copyright 2011 American Chemical Society.

Lin et al. [114] designed a gas sensor based on the LbL assembly of polyaniline prepared with polystyrene sulfonic acid as a template (PANI-PSSA) and titanium dioxide. This type of material presents a high sensitivity and fast response on the detection of $NH_3$, which is affected for the number of deposited bilayers. The origin of the good electrical properties of this type of material is associated with the formation of the p/n junction at the PANI-PSSA/titanium dioxide interface, and the high specific area of the material that facilitates the adsorption and diffusion of the ammonium molecules.

Easley et al. [115] designed electroactive coatings by the alternate deposition of layers formed by a polycation and a polyanion modified by pendant nitroxide groups. The obtained materials present an ion transport mechanism strongly dependent on the specific nature of the capping layer. In fact, films capped with a polyanion layer exhibit the highest capacity ($2\ \mu A/cm^2$) and a slightly reduced charge transfer resistance. Furthermore, the presence of the polyanion as the capping layer results in a more pronounced capacity fade than when the polycation is the capping layer.

Fang et al. [116] designed a novel capacitive bioanode for paper-based microbial fuel cells by combining conductive polypyrrole and bacterial cells. The electrochemical properties of this type of bioanode are excellent as a result of the combination of different effects: (i) Bacterial loading amount, (ii) conductivity, and (iii) metabolic activity loss during the fabrication process. Furthermore, the presence of polypyrrole, which plays the dual role of capacitive material and conductive spacers between bacterial layers, results in a good electron transfer efficiency, enhanced power density, and increased capacitance. Furthermore, the polypyrrole layers protect the bacterial layers, ensuring an efficient performance of the electrode.

Table 1 summarizes some examples of LbL nanostructures containing conductive polymers and their potential applications.

**Table 1.** Examples of LbL structures containing conductive polymers and their potential applications.

| System | Application | Reference |
| --- | --- | --- |
| MXene nanosheet and PNF | Supercapacitor electrodes | Yun et al. [108] |
| Polyaniline and Vanadium pentoxide | Capacitor electrodes | Shao et al. [109,110] |
| PEDOT and poly(N-methylpyrrole) | Thin film electrodes (anodes or cathodes) | Aradilla et al. [107] |
| PEDOT:PSS and polyethyleneimine | Capacitor electrodes | Lee et al. [111] |
| PEDOT:PSS and Carbon nanotubes | Capacitor electrodes | Agarwal et al. [112] |
| Polyaniline and MWCNT | Thin film electrodes | Hyder et al. [113] |
| PANI-PSSA and titanium dioxide particles | Gas sensors | Lin et al. [114] |
| Polycation and polyanion capped with nitroxide groups | Electroactive coatings | Easley et al. [115] |
| Polypyrrole and bacterial layers | Fuel cell electrode | Fang et al. [116] |

*3.2. LbL Electrochemical Devices Containing Carbon Nanomaterials*

Carbon nanomaterials are generally nanostructures based on $sp^2$ graphitic carbon atoms, e.g., fullerenes, carbon nanotubes, and graphene-like materials [117]. Their char-

acteristic electronic structure combined with their large specific area and a high chemical stability made them very interesting systems for the fabrication of different electrochemical devices [42].

Wang and Wang [118] proposed the fabrication of electrochemical capacitor electrodes for electric energy storage devices by the alternate deposition of negatively charged graphene nanosheets and a positively charged azo polymer using a dip-coating approach. The hierarchical structure of the assembled materials was formed by layers of densely packed graphene nanosheets which are interconnected along the 3D structure, forming random networks, with the azo polyelectrolyte playing an essential role in the prevention of the agglomeration of graphene nanosheets. The obtained LbL materials present a decreasing resistance with the increase of the number of bilayers up to a stationary value of $1.0 \times 10^6$ Ω after the deposition of 15 bilayers, which is three orders of magnitude lower than that corresponding to spin-coated films of the positively charged azo polymer. The low resistance and high accessible surface area of these materials make them a very interesting option for the fabrication of components for energy storage devices. It should be noted that the number of deposited bilayers is essential for controlling the quality of the obtained devices. In fact, the higher the number of bilayers the more continuous the charge transport pathways in the material. Furthermore, cyclic voltammetry showed the good capacitance of the LbL materials obtained by combining graphene nanosheets and azo polymers. These materials show enhanced capacitive properties with the increase of the number of deposited bilayers. Bulakhe et al. [119] also explored the use of graphene based nanomaterials for the fabrication of electrodes for energy storage devices. They combined layers of molybdenum sulfide and reduced graphene oxide nanosheets, obtaining multilayered structures with a high specific surface area. These structures present a good electrical conductivity with a specific capacitance of about 1070 F/g under a current density of 2 A/g, which is a 1.5-fold increase in relation to the performance of molybdenum sulfide electrodes.

Lee et al. [120] designed electrodes based in the alternate deposition of multiwalled carbon nanotubes (MWCNT) functionalized with carboxylic and ammonium groups capped with a $MnO_2$ layer obtained upon the treatment of the LbL material with potassium permanganate. This type of material offers a good volumetric capacitance (246 F/cm$^3$ at scan rate of 10 mV/s) and good cycling stability (up to 1000 cycles), which provides interesting properties for the manufacturing of electrical capacitors due to the good retention capacity (up to 1000 mV/s), emerging from the fast transport of ion and electron within the electrodes. Furthermore, the surface charge densities of the electrodes increases with the electrode thickness, which provides a very simple strategy to module the electrochemical response of the manufactured electrodes.

Zhou et al. [121] built flexible and foldable electrodes by the spray-assisted LbL coating of polycaprolactone fiber networks using alternate layers of MXene nanoflakes and MWCNT. The obtained materials combine polycaprolactone networks with MWCNT as spacers. This minimizes the restacking of the MXene flakes and increases the area accessible to the active materials, contributing to the overall charge storage and the high rate performance. In fact, the electrode structure favors a fast diffusion of the electrolyte ions within the electrodes, which results in a capacitance per area unit in the range 30–50 mF/cm$^2$, and a very good rate performance (14–16% capacitance retention at a scan rate of 100 V/s). On the other hand, the obtained electrodes present a very high resistance against repeated mechanical stresses. The assembly of alternated layers of MXene and carbon nanotubes was also exploited by Weng et al. [122] for the fabrication of lightweight, flexible, and electrically conductive thin films with application in electromagnetic interference shielding. The obtained composite films had high conductivity (up to 130 S/m) and high specific shield effectiveness (close to 60,000 dB·cm$^2$/g). These characteristics are among the highest reported in the literature, and possibly originated from the combination of the high conductivity of the assembled components and the layered structure provided by the LbL method.

Gittleson et al. [123] assembled LbL films by combining single-walled carbon nanotubes (SWCNT), and vanadium pentoxide nanowires or sodium poly(4-styrenesulfonate)

(PSS) with the aim of fabricating transparent anodes or cathodes, respectively. These electrodes present a good performance in Li-ion cells, exhibiting reversible lithiation capacities of 23 and 7 $\mu A \cdot h/cm^2$ for anodes and cathodes, respectively at a current density of $5 \mu A/cm^2$. The combination of both electrodes in a single fuel cell allows a performance at a capacity of about $5 \mu Ah/cm^2$ over 100 cycles. It is worth mentioning that the use of LbL films for the fabrication of components for lithium ion batteries also presents advantages from the mechanical point of view. This can be understood considering that one of the main limitations of the materials used as components in lithium ion batteries is associated with their mechanical failure due to their volume expansion and dissolution during lithiation/delithiation processes. However, the use of LbL structures can prevent this problem, minimizing the diffusion-induced stresses and preventing the electrode dissolution [124–127]. The fabrication of transparent electrodes by using carbon nanomaterials assembled by the LbL method was also explored by Camic et al. [128]. They designed transparent electrodes with a resistance of about 9 $\Omega$ by the alternate assembly of three bilayers formed by individual layers of graphene oxide and copper nanowires.

Neuber et al. [129] explored the effect of the molecular weight of poly(diallyldimethyla mmonium chloride) (PDADMAC) in the conductivity of LbL structures formed by the deposition of alternate layers of PDADMAC and oxidized carbon nanotubes. The obtained structures present an organization of the carbon nanotube layers with a surface parallel orientation, increasing its coverage with the number of deposited bilayers. On the other hand, the molecular weight of the polymer does not modify either the film thickness or the electrical conductivity of the material. The latter reaches values of up to $4 \times 10^3$ S/m after the deposition of four bilayers, and then there are no changes in the conductivity after the deposition of additional layers. This can be understood considering that a minimum number of layers is required for ensuring a constant coverage of carbon nanotube crossing point or nodes. Even though the molecular weight of the polymer does not modify the characteristics of the film, these can be modified by changing the nanotube concentration. The increase of this parameter increases the thickness of the film, which reduces the conductivity of the layers as a result of the increase of the monomer/carbon nanotube ratio. Therefore, it is possible to assume that the PDADMAC chains adsorb on the carbon nanotube surface, and it is the shape of the assembled carbon nanotube networks concerning the parameter that governs the thickness of the fabricated LbL material.

Shakir [130] fabricated thin film electrodes using the LbL assembly of two types of carbon nanomaterials: MWCNT and graphene nanosheets. The inclusion of conductive MWCNT spacers between graphene layers prevents the aggregation of the latter, increasing the number of sites available for the ion intercalation. However, the MWCNT also plays a very important role as conductive additive, spacer, and binder. This allows the assembly of materials with a high electrode effective surface area, making possible the fabrication of supercapacitors with a very high electrochemical capacitance (390 F/g) and excellent cycling stability after 25,000 charge–discharge cycles. Furthermore, the MCWCNT layer leads to the increase of both energy and power densities in relation to bare graphene electrodes (more than 30% increase).

Table 2 summarizes some examples of LbL nanostructures containing carbon nanomaterials and their potential applications.

**Table 2.** Examples of LbL structures containing carbon nanomaterials and their potential applications.

| System | Application | Reference |
|---|---|---|
| Graphene nanosheets and azo-polymer | Capacitor electrodes | Wang and Wang [118] |
| Reduced graphene oxide nanosheets and molybdenum sulfide | Capacitor electrodes | Bulakhe et al. [119] |
| Alternate layers of MWCNT modified with ammonium and carboxylic group capped with a layer of MnO$_2$ | Capacitor electrodes | Lee et al. [120] |
| MXene and MWCNT | Supercapacitor electrodes | Zhou et al. [121] |
| MXene and Carbon nanotubes | Thin films for electromagnetic interference shielding | Weng et al. [122] |
| SWCNT and PSS | Fuel cell anodes | Gittleson et al. [123] |
| Graphene oxide and copper nanowires | Transparent electrodes | Camic et al. [128] |
| PDADMAC and oxidized carbon nanotubes | Conductive layers for biological applications | Neuber et al. [129] |
| MWCNT and graphene | Thin film electrodes | Shakir [130] |

*3.3. LbL Electrochemical Devices Containing MXene*

MXene are a family of 2D materials, including transition metal carbides, nitrides, or cabonitrides with a general formula $M_{n+1}X_nT_x$, with M indicating an early transition metal (e.g., Ti, Zr, V, Nb, Ta, or Mo), X is carbon and/or nitrogen, and $T_x$ accounts for any functional groups on the MXene surface. MXene present physical, chemical, and biological properties similar to graphene-like materials. Furthermore, MXene contains different types of functional groups (-O, -OH or -F), which opens important avenues for the modification of their surface chemistry, and for their application as electrode materials in catalysis, batteries, supercapacitors, or sensors [131]. In fact, MXene have good electrical conductivity, large specific surface area and light weight and they are easy to process [132].

Above were discussed some examples in which MXene were used in electrochemical devices combined with carbon nanomaterials, but it is also possible to exploit MXene as the main electroactive compound. Tian et al. [133] assembled electrochemical devices containing MXene in combination with layers of a small molecule, such as tris(2-aminoethyl) amine. This allows the fabrication of a layered structure of MXene with an interlayer spacing of about 1 Å, which favors the interconnection between the electroactive MXene. This results in LbL materials with an extremely high electronic conductivity (above $7 \times 10^4$ S/m), allowing their use as electrodes for flexible all-solid-state supercapacitors. These deliver a high volumetric capacitance (583 F/cm$^3$), and high energy (3 W·h/L) and power density (4400 W/L), emerging as a very interesting alternative for the fabrication of a broad range of highly conductive structures. Figure 7 provides two sketches representing the structures of MXene-tris(2-aminoethyl) amine layers deposited in templates allowing the fabrication of fiber and 3D porous structure. Furthermore, a set of images obtained using Electronic Scanning Microscopy of the LbL films on different types of substrates is also shown.

Jin et al. [134] manufactured multifunctional hybrid materials composed by MXene and poly(vinylalcohol) layers. These materials present a structure composed by continuous MXene network embedded in a poly(vinylalcohol) gel favoring good heat and electron conduction, and simultaneouly have excellent electromagnetic interference shielding efficiency and thermal conductivity. Both the electrical conductivity of the materials (up to 716 S/cm) and the electromagnetic interference shielding appear strongly dependent on the multilayer thickness. In particular, the latter parameter presents its maximum value for LbL films of 25 μm (44.4 dB). Therefore, the alternation of conductive and non-conductive layers allows the fabrication of materials with a strong internal scattering and absorption of electromagnetic radiation.

Lipton et al. [135] designed multifunctional free-standing films formed by 2D titanium carbide nanosheets (MXene) and clay nanoplatelets, which present a thickness dependent tensile strength (in the range 138–225 MPa), a relatively high electromagnetic interference shielding (up to 24,550 dB·cm$^2$/g), and conductivities in the range 53–125 S/m. This behavior arrives from the nacre-like brick-and-mortar structure, allowing applications that range from the fabrication of membranes to the design of structural and multifunctional composites, and from energy harvesting and storage materials to materials for aerospace.

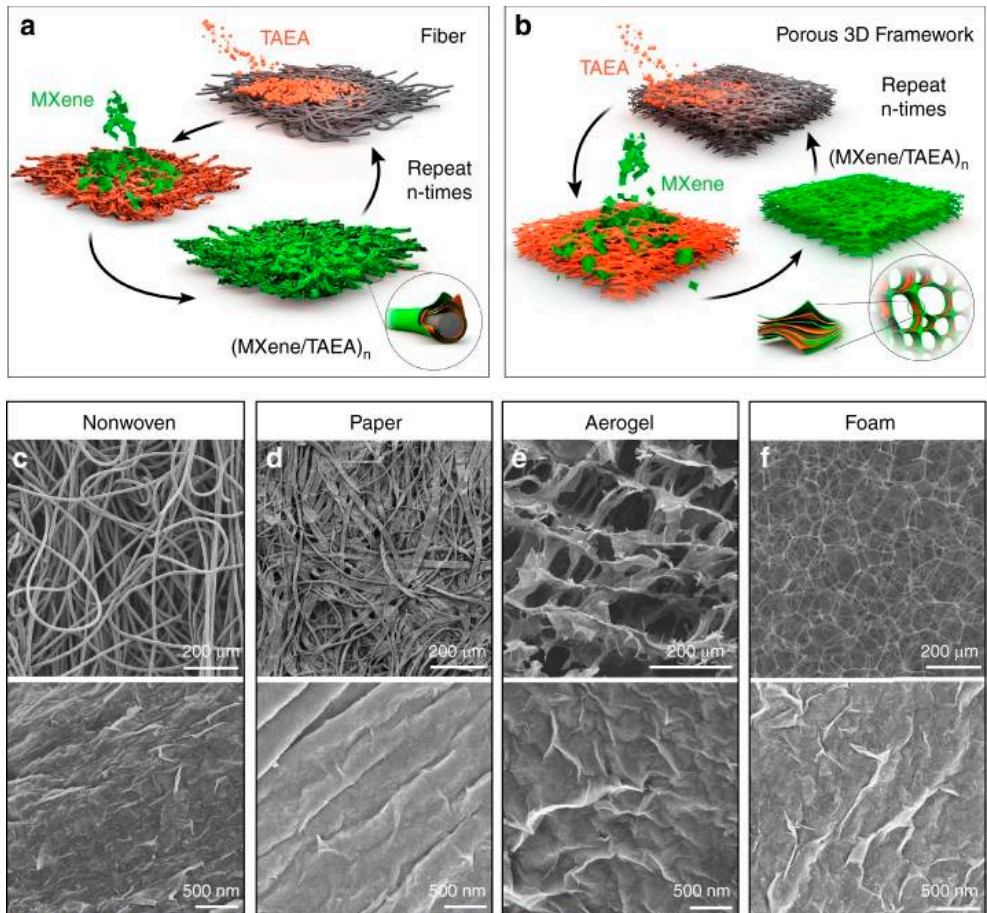

**Figure 7.** Sketch of the assembly process of LbL multilayers obtained by alternate deposition of MXene and tris(2-aminoethyl) amine for fiber (**a**) and 3D porous structures (**b**), and a set of images at two different magnifications of the LbL structures on different templates: Non-woven template (**c**), paper (**d**), aerogel (**e**) and foam (**f**). Reprinted from Tian et al. [133], with permission under Open access CC BY 4.0 license, https://creativecommons.org/licenses/by/4.0/ (accessed on 14 April 2022).

Liu et al. [136] designed LbL films on silk fabrics by combining MXene and silver nanowires. These present a leaf-like conformation where MXene works as the conductive layer and the silver nanowires as the conductive veins (see Figure 8a for the example of the building process and Figure 8b for the final structure and morphology of the assembled supramolecular systems). The increase of the concentration of the latter reduces linearly the conductivity of the films and their air permeability. On the other hand, the ability of the films to provide electromagnetic interference shielding is enhanced with the number of deposited bilayers and the amount of silver (see Figure 8c for the evolution of the electromagnetic interference shielding capacity with the number of bilayers, n). For instance, films with thickness of 120 μm present an electromagnetic interference shielding of 54 dB in the X-band frequency range. Figure 8d represents the electromagnetic interference shielding performance and the film resistance for LbL materials obtained using different fabrics as substrate. It should be noted that materials can be used as highly sensitive humidity sensors, retaining their porosity and permeability.

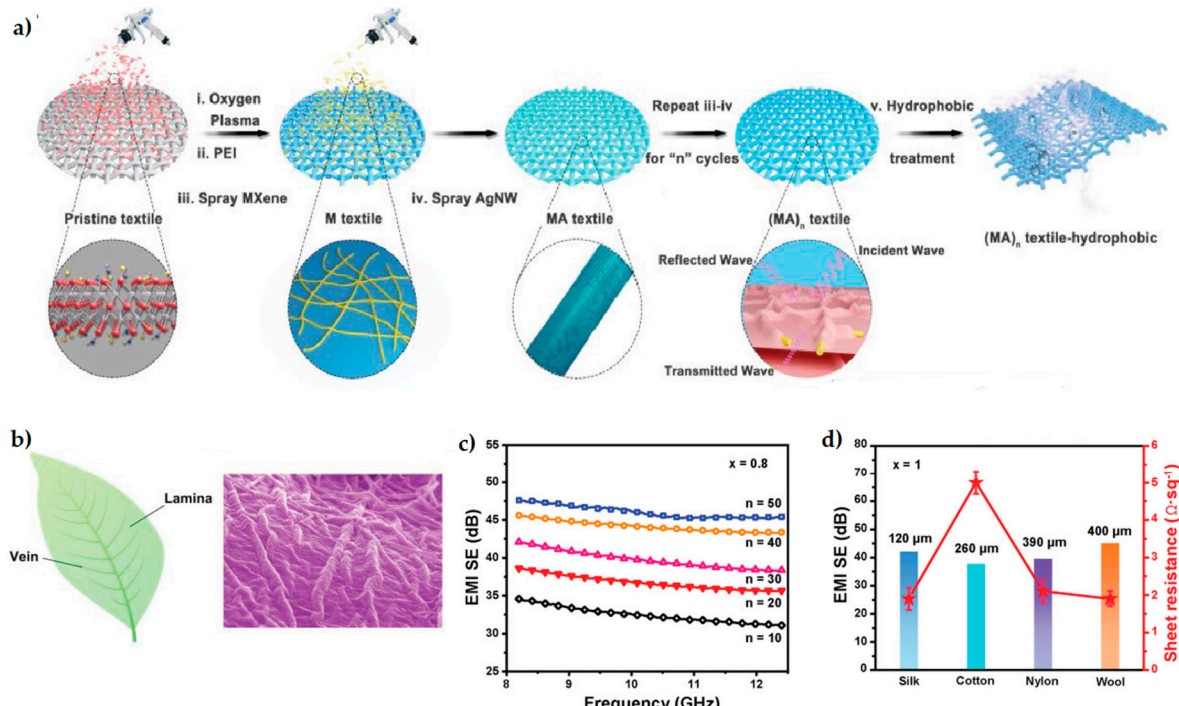

**Figure 8.** (**a**) Sketch of the assembly process of LbL multilayers obtained by the alternate deposition of MXene and silver nanowire layers. Notice that before the deposition of the multilayered structure on the plasma treated substrate a polyethyleneimine (PEI) layer was deposited. (**b**) Structure and morphology of the LbL films obtained by the alternate deposition of MXene and silver nanowire layers. (**c**) Effect of the number of deposited bilayers on the electromagnetic interference shielding. (**d**) Electromagnetic interference shielding (bar chart, left axis) and sheet resistance (red symbols, right axis) for LbL films using different fabrics as substrates. The thicknesses of the LbL films are also included within the panel. Reprinted with permission from Ref. [136]. Copyright 2019 John Wiley and Sons.

An et al. [137] manufactured humidity sensors by combining layers of MXene and PDADMAC. These sensors work by changing the separation distance between MXene layers as a result of the swelling/shrinking of the PDADMAC layers when the environmental humidity changes. This induces a modification of the tunneling resistance between MXene sheets, allowing the detection of humidity changes in a reversible way. Figure 9 shows the change of the ratio between the tunneling resistance at a fixed value of the relative humidity and tunneling resistance at a relative humidity of 10%, and the change of conformation as result of the changes of humidity.

Most of the studies dealing with MXene layers consider the use of $Ti_3C_2T_z$. However, Echols et al. [138] explored the assembly of two different MXene, $Ti_2CT_z$ and $Nb_2CT_z$, with PDADMAC layers. This study demonstrated that both the type of metal and the stoichiometry of the MXene present a very critical role on the growth and properties of the LbL material. For instance, the higher the metal content on the MXene the higher its conductivity and stability against oxidation. On the other hand, the use of MXene containing Nb reduced the band gap in relation to MXene containing Ti, and hence the conductivity of the assembled materials is favored.

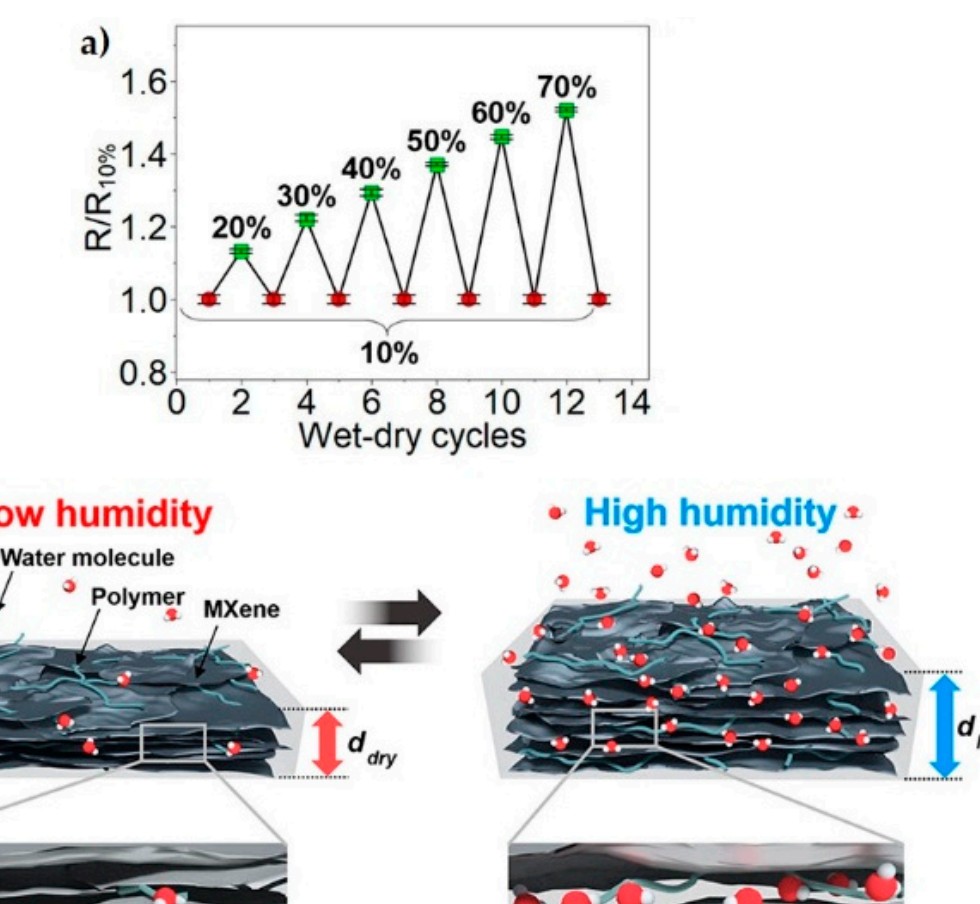

**Figure 9.** (**a**) Ratio between the tunneling resistance at a fixed value of the relative humidity (indicated in the plot) and the tunneling resistance at a relative humidity of 10%. (**b**) Sketch of the proposed response to the humidity of the MXene-PDADMAC films. Reprinted with permission from Ref. [137]. Copyright 2019 American Chemical Society.

Table 3 summarizes some examples of LbL nanostructures containing MXene and their potential applications.

**Table 3.** Examples of LbL structures containing MXene and their potential applications.

| System | Application | Reference |
|---|---|---|
| MXene and tris(2-aminoethyl) amine | Supercapacitor electrodes | Tian et al. [133] |
| MXene and poly(vinyl alcohol) | Electromagnetic interference shielding devices | Jin et al. [134] |
| MXene and nanoclays | Multifunctional materials | Lipton et al. [135] |
| MXene and silver nanowires | Electromagnetic interference shielding devices and Humidity sensors | Liu et al. [136] |
| MXene and PDADMAC | Humidity sensors | An et al. [137] |

It should be stressed that the use of the LbL methods in the fabrication of MXene-based nanomaterials allows the assembly materials with density, electronic properties, and mechanical strength to be tuned almost at will. This allows fabricating materials with a

broad range of electromagnetic interference shielding efficiency, which makes it possible to fulfill specific standards.

### 3.4. LbL Electrochemical Devices Containing Pseudocapacitive Nanoparticles

Pseudocapacitive particles are commonly obtained from oxides and hydroxides of transition metals, which are characterized by the emergence of a rapid charge transfer of surface-bound species. The use of pseudocapacitive materials allows overcoming some of the main kinetics limitations found in conventional materials obtaining devices with very high energy densities during their performance [139].

Song et al. [140] designed energy storage electrodes by the alternate assembly of pseudocapacitive iron oxide ($Fe_3O_4$) nanoparticles and conductive indium tin oxide (ITO) ones by the LbL method. For this purpose, it was necessary to modify the particles with different organic ligands to tune the charge-transfer efficiency. For instance, the capping of the particles with hydrazine ligands significantly enhances the performance of the fabricated electrodes by reducing the separation between adjacent particles and increasing the number of oxygen vacancies, which increases the rate capability and capacitance of the electrodes. This contributes to an enhanced performance in energy storage.

Sharma et al. [141] designed supercapacitor electrodes by the alternate deposition of $Sr(OH)_2$ and $CoO(OH)$ layers on stainless steel substrate. The obtained electrodes for this method present a relatively high specific capacitance (around 1550 F/g), and a good stability over 12,000 charge–discharge cycles at 8 mA (~67%). This allows the fabrication of solid state symmetric supercapacitors with a specific capacitance of 155.2 F/g and a stability of about the 70% over 6000 operation cycles at 3 mA current rate, providing high energy (49.9 W·h/kg) and power (7088.5 W/kg) densities. This allows its application for illuminating light emitting diodes of different colors.

Nam et al. [142] assembled alternate layers of $LiFePO_4$ and conductive ITO nanoparticles onto aluminum textiles for fabricating lithium-ion battery cathodes. These present high areal capacity and rate performance, and good cycling stability. This type of structure favors an easy charge transfer within the manufactured electrodes.

Yun et al. [143] designed electrochromic films by combining alternating layers of oleylamine stabilized tungsten oxide nanorods and oleylamine stabilized ITO nanoparticles, adding tris(2-aminoethyl)amine as the linker. This allows the fabrication of supramolecular architectures where the distance between conductive layers is minimized, resulting in a periodic insertion of transparent/conductive ITO nanoparticles within the LbL film. This reduces the charge transfer resistance associated with the presence of tungsten oxide nanorods, improving the electrochemical performance of the assembled nanomaterials. The manufactured electrochemical films present a faster switching response than devices without ITO, and a maximum optical modulation of about 56%. Choi et al. [144] also used molecular linkers to connect alternate layers of manganese oxide (MnO) and ITO nanoparticles, which alows a significant reduction of the separation between pseudocapacitive nanoparticles. Furthermore, the presence of ITO layers between those of manganese oxide contributes to a reduction of the resistance associated with the transference of charge, improving the capacitive performance of the devices. In fact, the intercalation of ITO layers between MnO ones results in a twofold increase of the areal capacitance in relation to bare MnO electrodes.

Table 4 summarizes some examples of LbL nanostructures containing pseudocapacitive nanoparticles and their potential applications.

**Table 4.** Examples of LbL structures containing pseudocapacitive nanoparticles and their potential applications.

| System | Application | Reference |
|---|---|---|
| $Fe_3O_4$ and ITO nanoparticles | Capacitor electrodes | Song et al. [140] |
| $Sr(OH)_2$ and $CoO(OH)$ nanoflakes | Supercapacitor electrodes | Sharma et al. [141] |
| ITO nanoparticles and $LiFePO_4$ | Lithium-ion battery cathodes | Nam et al. [142] |
| Tungsten oxide nanorods and ITO nanoparticles | Electrochromic films | Yun et al. [143] |
| Manganese oxide and ITO nanoparticles | Supercapacitor electrodes | Choi et al. [144] |

*3.5. LbL Electrochemical Devices Containing Layered Double Hydroxides*

Layered double hydroxides are inorganic materials with a layered structure defined for a general molecular formula $\left[M_{1-x}^{II}M_x^{III}(OH)_2\right]^{z+}(A^{n-})_{z/n} \cdot yH_2O$, with $M^{II}$ and $M^{III}$ representing divalent (e.g., Mg, Ni, Co, Zn or Fe) and trivalent (Al, Fe, Cr or Ga) cations, respectively, and $A^{n-}$ is the interlayer anion compensating the charge of the cationic layers. The composition of layered double hydroxides can be easily tuned by changing the nature of the metal cations and the anionic counterions without modifying their structure, which open very interesting possibilities for their uses in catalysis, adsorption, biology, energy storage, and conversion. Furthermore, the elevated dispersion of active species within the layered matrix makes possible the exfoliation of the layered double hydroxide into monolayer sheets, and their functionalization allowing their exploitation in the fabrication of multifunctional electrodes. In fact, different electrodes have been fabricated by assembly positively charged layered double hydroxide nanosheets and negatively charged species [145]. Shao et al. [146] fabricated LbL materials by combining layers of iron (III) porphyrin and layered double hydroxide nanosheets of cobalt and aluminum. This type of electrode presents two well-defined reversible redox peaks associated with the Co(III)/Co(II) couple, whereas iron (III) porphyrin facilitates the electron transfer. On the other hand, the manufactured electrodes present an excellent electrocatalytic activity for $H_2O_2$, ensuring a large range of linear response, high sensitivity, and very low detection limit. Zhang et al. [147] also combined layered double hydroxide of cobalt and nickel and iron (III) porphyrin. In this case, they exploited this system on the fabrication of ultrathin film electrodes which present a very low overpotential (264 mV) to obtain current density for the oxygen evolution reaction of 10 $mA/cm^2$, and a decrease of the Tafel slope of 37.6 mV/dec.

A very popular alternative for the assembly of layered double hydroxides in LbL materials is their combinations with functional molecules, including naphthol green B, cobalt phthalocyanine, ruthenium(II) complex, or biprotein. This allows the fabrication of nanomaterials with very promising electrochemical properties [145]. Kong et al. [148] designed a hydrogen peroxide electrochemical sensor by combining layers of anionic napthol green B and a cationic double layered hydroxide of cobalt and aluminum. The obtained sensor shows a fast electron transfer based on the Co(III)/Co(II) redox couple, resulting in an electromotive force of 14 mV. Furthermore, the obtained electrodes modified with the LbL film display a significant electrocatalytic activity for $H_2O_2$, presenting a rapid response, high stability and selectivity, and good reproducibility. Zhang et al. [149] designed a solid-state electrochemiluminescence sensor for dihydronicotinamide adenine dinucleotide formed by a sequence of layers composed of layered double hydroxide-PSS-ruthenium(II) complex $(Ru(bpy)_3^{2+})$-PSS-layered double hydroxide. This sensor presents a liner response of the detection of the dinucleotide with a relative low detection limit of 0.023 μM. The fabrication of electrochemiluminescence systems by using layered double hydroxide was also explored by Li et al. [150]. They assembled alternate layers of layered double hydroxide of cobalt and aluminum capped with luminol and anionic CdTe quantum dots, observing the existence of an effective electrochemiluminescence resonance energy transfer from luminol (donor) to quantum dots (acceptor). Furthermore, the electrochemiluminescence intensities of both luminol and quantum dots undergo a simultaneous enhancement as a result of

the sensitization of the quantum dots and the occurrence of the resonant process. This allows the fabrication of electrodes with good sensitivity and selectivity for the detection of trinitrotoluene.

Han et al. [151] designed a dopamine sensor by the LbL assembly of cobalt phthalocyanine and layered double hydroxide on ITO electrodes. The detection of the dopamine is possible by its oxidation in the presence of the Co(II)/Co(III) redox couple, resulting in high sensitivity, low detection limit, and reduced level of interferences.

Table 5 summarizes some examples of LbL nanostructures containing layered double hydroxides and their potential applications.

**Table 5.** Examples of LbL structures containing layered double hydroxides and their potential applications.

| System | Application | Reference |
| --- | --- | --- |
| Layered double hydroxide of cobalt and aluminum and iron (III) porphyrin | Electrodes for electrocatalysis | Shao et al. [146] |
| Layered double hydroxide of cobalt and aluminum and naphthol green B | Electrodes for electrocatalysis | Kong et al. [148] |
| Layered double hydroxide, PSS and Ru(bpy)$_3^{2+}$ | Electrochemiluminescence sensor | Zhang et al. [149] |
| Layer double hydroxide of cobalt and aluminum capped with luminol and anionic quantum dots of CdSe | Electrochemiluminescence sensor | Li et al. [150] |
| Layer double hydroxide and cobalt phthalocyanine | Dopamine sensor | Han et al. [151] |

*3.6. LbL Electrochemical Devices Containing Polyoxometalates*

Polyoxometalates (POMs) are anionic metal-oxides containing early transition metals (e.g., vanadium, niobium, tantalum, molybdenum, and tungsten) and trace amounts of heteroatoms (e.g., phosphorus, arsenic, silicon, or germanium). POMs present cluster-like structures characterized by their different shapes and sizes, and can be exploited as a result of their high thermal stability, high negative charges, remarkable redox abilities, unique ligand properties, and availability of organic grafting in a broad range of technological fields, e.g., medicine, catalysis, and electrochemistry [152]. The assembly of LbL electrochemical devices containing polyoxometalates provides a very favorable environment for electron transfer, opening new avenues for the design of sensors, microelectronics devices, or electrocatalytic systems [153]. In fact, LbL can help concerning the accurate regulation of POM loading, which is essential for improving the practical application of POM-based devices [152].

Genovese et al. [153] demonstrated that the assembly of electrodes formed by alternate layers of POMs and multi-walled carbon nanotubes (MWCNTs) on carbon surfaces provides the bases for the fabrication of devices having volumetric capacitances about 10 times larger than that expected for conventional electric double layer capacitors. Akter et al. [154] demonstrated that alternate layers of POM and MWCNT result in a synergistic effect on capacitance and kinetics, allowing the fabrication of powerful pseudocapacitive systems for energy storage. Similar synergistic effects were found in devices obtained upon the assembly by LbL of mesoporous $SnO_2$, gold nanoparticles, and a POM ($P_2W_{18}$) on Indium Tin Oxide (ITO) surface. This synergy results in a significant increase of the speed associated with electron transference. Furthermore, this type of electrochemical device presents an excellent selectivity, high reproducibility, and excellent stability [155].

The pseudocapacitive behavior of POM-based LbL devices is not only dependent on the specific architecture of the assembled nanomaterial, as demonstrated Park et al. [156]. They showed that the nature of the carbon material used as a template for the assembly of alternate layers of anionic Keggin-type polyphosphomolybdate and a traditional polycation, such as PDADMAC, presents a strong influence in the current response and the capacitance of the assembled electrochemical devices. This is associated with the different surface structure of the substrate which can enhance the conductivity and connectivity between the support and the POM units. For instance, the formation of a locally connected network structure in the association of onion-like carbon allows a strong increase of the capacitance

of the electrochemical material, reaching values of up to 600 mF/cm$^2$ at 5 V/s. Similar results are found when carbon nanotubes are used as substrates. However, the use of carbon nanodiamonds worsens the performance of the electrochemical devices. Therefore, the surface chemistry, structure, and hybridization of the carbon atoms (sp$^2$/sp$^3$) of the substrate presents a key role in the capacitance of the manufactured electrochemical devices.

Salimi et al. [157] demonstrated that the increase of the number of deposited layers on hybrid systems formed by POM and a complex of Ruthenium increases the current of the cathodic peak, allowing an effective control of the electrochemical properties of the LbL devices. This appears strongly dependent on the chemistry of the specific polyoxometalate used in the assembled material [158]. For instance, the modification of glassy carbon electrodes with alternate layers of a POM, polyethyleneimine, and polypyrrol enhances the electrochemical response and sensitivity of the bare electrode. This type of LbL modified electrode has a strong electrocatalytic activity for rapid detection of tyrosinase in aquatic medium [159].

Xu et al. [160] showed that LbL nanomaterials containing polyoxometalates can be exploited for the fabrication of electrochromic devices. They combined a polyoxometalate of tungsten with poly(hexyl viologen) in the fabrication of LbL materials which can undergo a reversible transition from colorless to violet by the progressive modification of the applied potential between 0.1 and −0.9 V.

Table 6 summarizes some examples of LbL nanostructures containing polyoxometalates and their potential applications.

**Table 6.** Examples of LbL structures containing polyoxometalates and their potential applications.

| System | Application | Reference |
|---|---|---|
| Polyoxometalates and MWCNT | Double layer capacitors | Genovese et al. [153] |
| Polyoxometalates, mesoporous SnO$_2$ and gold nanoparticles | Pseudocapacitive systems | Akter et al. [154] |
| Polyoxometalates and different carbon materials | Capacitors | Park et al. [156] |
| Polyoxometalates and complex of Ruthenium | Electrodes | Salimi et al. [157] |
| Polyoxometalate of tungsten and poly(hexyl viologen) | Electrochromic devices | Xu et al. [160] |

## 4. Conclusions

The new avenues opened for the Layer-by-Layer method on the processing of materials have driven its progressive introduction in the research, development, and innovation strategy of the technological world. The broad interest on the LbL method has also reached the frontiers of the fabrication of nanostructured materials with electrochemical purposes, and in particular those aimed for electrochemical energy storage and conversion. From the early 2000s, the research on LbL materials for the manufacturing of electrochemical devices by using different building blocks (conductive polymers, carbon nanomaterials, metal oxides/hydroxides, layered double hydroxides, or polyoxometalates) has blossomed, trying to solve many practical problems of conventional systems, attracting the attention of the research community. The different examples reviewed in this work evidence that the particular physical, chemical, mechanical, and electronic properties that can be attained by the assembly of LbL nanomaterials can provide the bases for the fabrication of novel miniaturized devices with enhanced electrochemical performance (conductivity, capacitance, etc.) combined with good mechanical properties. However, this combination of properties requires a careful design at the nanoscale of materials in which their structure and composition can guarantee an optimal performance, favoring an efficient electronic and ionic transfer through interconnected conducting networks. This makes of the LbL method a key enabling technology, facilitating the integration at the molecular level of the different building blocks on the structure of the electrode support. The possibilities offered for the integrative concept provided by the LbL method lead to the emergence of synergistic effects between the building blocks of the complex supramolecular systems, which enhances their performance in relation to the individual components.

It should be noted that the examples discussed in this review reflect some of the most promising facets of the molecular design of supramolecular systems for electrochemical applications obtained by using the LbL method, evidencing that the fabrication of LbL electrochemical devices should be analyzed as an open-ended story. This requires further efforts to ensure the manufacturing of more efficient and sensitive materials for electrochemical applications. Therefore, this review has tried to offer an updated perspective on how the integrative supramolecular approach offered by the LbL method can help in the solution of important problems related to electrochemistry.

**Funding:** This work was funded by MICINN under grant PID2019-106557GB-C21, and by E.U. on the framework of the European Innovative Training Network-Marie Sklodowska-Curie Action NanoPaInt (grant agreement 955612).

**Institutional Review Board Statement:** Not applicable.

**Informed Consent Statement:** Not applicable.

**Data Availability Statement:** This manuscript does not contain any associated data.

**Conflicts of Interest:** The authors declare no conflict of interest. The funders had no role in the design of the study; in the collection, analyses, or interpretation of data; in the writing of the manuscript, or in the decision to publish the results.

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
