# Peer review of "Layer-by-Layer Materials for the Fabrication of Devices with Electrochemical Applications"

_energies, doi:10.3390/en15093399_

Round 1

Reviewer 1 Report

The work is quite relevant and can be considered for publication. However, I recommend the observation of the following points.

1) I recommend replacing Fig. 1 for a more creative one.

2) In Fig. 2, it is necessary to clarify what the blue and red lines represent.

3) It is essential that authors carefully examine the writing of the text. There are several types of errors, for example, see lines 151, 153, 158, 244, 333, 471, 496, 498, 509, 557 and 692.

4)  In my opinion, the structure of the work needs improvement, for example, sections 4-9 are actually subsections of section 3.

5) Explain the meaning of "PEI" in Fig. 8a.

6) In particular, I consider the use of the term "electrochemical charge transfer" technically inappropriate (L. 642). 

Author Response

The work is quite relevant and can be considered for publication. However, I recommend the observation of the following points.

  • I recommend replacing Fig. 1 for a more creative one.

We thank the reviewer recommendation. However, after analyzing the figure, we have decided to maintain it because it provides all the required information in a very clear way.

2) In Fig. 2, it is necessary to clarify what the blue and red lines represent.

The meaning of the blue and red lines have been highlighted in the figure legend.

3) It is essential that authors carefully examine the writing of the text. There are several types of errors, for example, see lines 151, 153, 158, 244, 333, 471, 496, 498, 509, 557 and 692.

We have revised the text.

4)  In my opinion, the structure of the work needs improvement, for example, sections 4-9 are actually subsections of section 3.

Following the reviewer recommendation, we have converted the sections 4 to 9 in subsections of section 3.

5) Explain the meaning of "PEI" in Fig. 8a.

We have included the meaning of PEI in the Legend of the figure.

6) In particular, I consider the use of the term "electrochemical charge transfer" technically inappropriate (L. 642).

We have modified the mentioned term.

We thank to the reviewer for the comments, they were very useful for improving the quality of our manuscript. 

Reviewer 2 Report

This work provides an updated perspective to the current trends on the use of LbL materials for the fabrication of materials for electrochemical applications, highlighting some of the most recent research efforts in this field. This work seems to be interesting and could be recommended for publication after some minor revisions.

  1. For the LBL materials in the lithium ion batteries, some discussions on the lithiation deformation should be provided in order to explain their advantages in the LIB applications, such as: Journal of Electroanalytical Chemistry 767 (2016) 49-55; Journal of Power Sources 290 (2015) 114-122; Journal of The Electrochemical Society 163 (2016) A1157-A1163; Extreme Mechanics Letters 9 (2016) 226-236.
  2. In Section 3, some experimental data should be provided.
  3. In Section 5, it is better if the authors could separate the carbon materials as particles, wires, and tubes etc, with measuring pictures.

Author Response

This work provides an updated perspective to the current trends on the use of LbL materials for the fabrication of materials for electrochemical applications, highlighting some of the most recent research efforts in this field. This work seems to be interesting and could be recommended for publication after some minor revisions.

  1. For the LBL materials in the lithium ion batteries, some discussions on the lithiation deformation should be provided in order to explain their advantages in the LIB applications, such as: Journal of Electroanalytical Chemistry 767 (2016) 49-55; Journal of Power Sources 290 (2015) 114-122; Journal of The Electrochemical Society 163 (2016) A1157-A1163; Extreme Mechanics Letters 9 (2016) 226-236.

We have added a brief comment about the problem raised by the reviewer to the text.

  1. In Section 3, some experimental data should be provided.

The experimental data are included in the subsections of Section 3, to introduce any additional data in the section will be redundant.

  1. In Section 5, it is better if the authors could separate the carbon materials as particles, wires, and tubes etc, with measuring pictures.

The information existing in the literature is not enough to perform a detailed discussion as individual categories for carbon materials, and hence this separation on the manuscript does not provide any additional insight or improvement.

We thank to the reviewer for the comments, they were very useful for improving the quality of our manuscript. 

Round 2

Reviewer 1 Report

1) L. 177: The author's correct name is Guzmán.

2) L. 342: The text refers to Fig. 5 and not Fig. 4.

3) L. 655: Should be changed to "band gap".

4) L. 824 and Table 5: Should be changed to "hexyl viologen".

5) L. 1189: Remove the asterisk.

6) Please check English language, for example, L. 733 "This type of electrodes present..." and L. 805 "For instance*,*"

Author Response

  • 177: The author's correct name is Guzmán.

We have corrected.

  • 342: The text refers to Fig. 5 and not Fig. 4.

We have corrected

  • 655: Should be changed to "band gap".

We have corrected.

  • 824 and Table 5: Should be changed to "hexyl viologen".

We have corrected.

  • 1189: Remove the asterisk.

We have corrected.

  • Please check English language, for example, L. 733 "This type of electrodes present..." and L. 805 "For instance*,*"

We have checked

We thank to the reviewer for the comments, they were very useful for improving our manuscript.